# Learning a Bi-directional Driving Data Generator via Large Multi-modal Model Tuning

## Abstract

Understanding human driving behaviors is crucial for developing a reliable vehicle and transportation system. Yet, data for learning these behaviors is scarce and must be carefully labeled with events, causes, and consequences. Such data may be more difficult to obtain in rare driving domains, such as in high-performance multi-car racing. While large language models (LLMs) show promise in interpreting driving behaviors, the integration of multi-modal inputs (e.g., language, trajectory, and more) and generation of multi-modal output in low-data regimes remains under-explored. In this paper, we introduce Bi-Gen: a **Bi**-directional Driving Data **Gen**erator, Bi-Gen is a bi-directional multi-modal model that connects a trained encoder-decoder architecture with a pre-trained LLM, enabling both auto-annotation and generation of human driving behaviors. Our experiments show that Bi-Gen, despite its smaller size, matches the performance of much larger models like GPT-4o in annotating driving data. Additionally, Bi-Gen generates diverse, human-like driving behaviors, offering a valuable tool for synthetic data generation in resource-constrained settings. Taken together, our experiments are a significant step towards applying LLMs to complex, multi-agent driving data.

## 1 Introduction

Large language models (LLMs) and large multi-modal models (LMMs) have emerged as capable and general-purpose tools for understanding driving data in the wild (Kuo et al., 2022; Felemban et al., 2024; Li et al., 2024b; Chen et al., 2023b; Xu et al., 2024; Sima et al., 2023). However, the key ingredient to the success of such models is the vast amounts of data required to pre-train or fine-tune such models to contain relevant world knowledge for target tasks. This data dependency becomes a significant limiting factor when extending LLMs to driving data that is not well-represented in publicly available datasets. For example, most empirical human driving data is naturalistic, which inherently biases it against capturing rare events (e.g., safety critical scenarios, drifting, etc.). Though synthesizing data using driving simulators or applying variance reduction techniques (e.g. importance sampling, etc.) could augment or extend existing datasets under specific scenarios (Feng et al., 2023; Ding et al., 2023), these approaches are inadequate for generating a diverse range of driving behaviors that capture the implicit heterogeneity of human driving. Furthermore, while some machine learning models (Wang et al., 2019; Krajewski et al., 2018; Huang et al., 2020; Phan-Minh et al., 2020; Nayakanti et al., 2023) may function as synthetic data generators, such methods often require significant amounts of labeled data to learn to generate realistic in-distribution examples.

To this end, we aim to to develop an efficient, low-cost multi-modal model that can quickly learn to interpret and annotate unlabeled driving trajectories in the low-data domain of high-performance multi-car racing (Weiss & Behl, 2020; Wurman et al., 2022; Chen et al., 2023a; Werner et al., 2023). Learning such a model involves two core tasks: (1) trajectory generation and (2) trajectory description. Trajectory generation involves the creation of new driving trajectories, either conditioned on a language prompt, a partially-complete trajectory, or some other form of driving context. Trajectory description is the annotation of an unlabeled driving trajectory, giving the model the ability to act as an annotator or discriminator for unlabeled data. One promising method is to extend LLMs for the two tasks, as such models have been shown to successfully serve as both synthetic data generators for training large-scale models (Dubey et al., 2024; Adler et al., 2024) and as annotators for unstructured data (Tan et al., 2024).

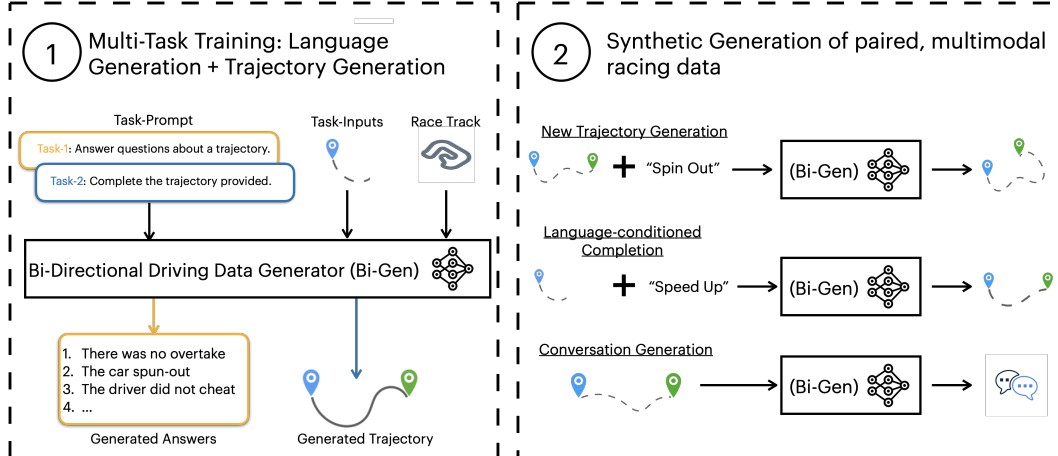

Figure 1: Our data generation framework, Bi-Gen, is trained as a multi-task, multi-modal generative model. (1) Bi-Gen model is trained to both annotate unlabeled trajectories in a multi-turn conversations and to generate completions to partial trajectories given language prompting. (2) At test-time, Bi-Gen can serve to convert existing trajectories into new variations, complete partial trajectories in accordance with a language prompt, or annotate unlabeled data in a multi-turn conversation.

The community has seen a growing interest in leveraging LLMs (e.g., GPT-4 (Achiam et al., 2023), ChatGPT (Biswas, 2023), etc.) for trajectory generation, particularly for automated vehicles (AVs) (Chen et al., 2023b; Lan et al., 2024; Nguyen et al., 2024; Cui et al., 2024a; Zhang et al., 2024c; Tan et al., 2023; Munir et al., 2024). These works demonstrate a strong capability of LLMs to directly translate high-level textual driving commands to lower-level trajectory data in the form of generated way-points, or by aligning LLM outputs to a vehicle's action space (e.g., steering, acceleration). Pre-training a decoder to align the LLMs' hidden-state to driving state vectors (Chen et al., 2023b; Mao et al., 2023) is effective for generating immediate actions or reasoning within specific time steps but lacks flexibility for scaling to varying time horizons. This task is complicated further by the need to understand temporal and spatial interactions across multiple modalities and multiple agents in a scene. Modeling such interactions requires dedicated fine-tuning of a pre-trained LLM, and in particular requires sufficiently rich multi-agent data.

Efficient trajectory generation demands a comprehensive representation of the complex driving environment, which necessitates the integration of multi-modal inputs. Challenges arise in establishing multi-modal connections within LLMs, in terms of the representation gap across modalities, cross-modality generalization, modality collapse, etc. (Yin et al., 2023; Zhang et al., 2024a; Peng et al., 2023; Driess et al., 2023; Ye et al., 2024). Particularly in the human-driving domain, generation is more complicated due to the scarcity of large-scale labeled trajectory-language paired datasets that incorporate multiple modalities (Tan et al., 2023; Cui et al., 2024b; Shao et al., 2024). These complexities are further amplified in the uncharted domain of high-performance multi-agent racing, which features complex multi-agent dynamics, no pre-trained data encoders, and no readily accessible world-knowledge baked into existing LLMs.

Finally, these methods only enable one-directional generation, focusing exclusively on language-conditioned trajectory generation. Equally important, however, is the reverse process: trajectory-conditioned language generation. The ability to generate multi-modal outputs (i.e., language and trajectory) from multi-modal inputs is essential for achieving bi-directionality in data synthesis. While prior works have individually tackled one element of this process, a bi-directional model capable of mutual conditional generation between language and trajectory is key to gaining a comprehensive understanding of human driving dynamics. Without this, the model is limited in generalizing across different driving contexts and behaviors, as it cannot fully explain or generate the diverse factors that shape and influence various driving behaviors.

To address these gaps, we introduce a **Bi**-directional Driving Data **Gen**erator, Bi-Gen, an end-to-end learning framework developed through LMM tuning. This bi-directional pipeline is capable

of simultaneously handling both trajectory description and trajectory generation, which is particularly important for low-data domains. We provide an overview of our framework in Fig. 1. Our contributions are summarized as follows:

- *Large multi-modal model for human driving comprehension*: We develop a robust LMM which effectively handles multi-modal inputs and outputs. Our experiments validate the effectiveness of the model in comprehending diverse human driving behaviors in a complex driving environment with limited training data.

- *Bi-directional trajectory-language interaction pipeline*: We present a multi-modal interaction enabling both language-conditioned trajectory generation and trajectory-conditioned language generation. We specifically show how this interaction processes trajectory inputs, describes trajectories using natural language, and reconstructs or generates diverse trajectories conditioned on the language inputs. This enables a human user to actively query about unlabeled data or to generate synthetic datasets to supplement small real data.

- *Synthetic data generator*: We further validate that Bi-Gen can serve as a synthetic data generator and annotator on par with closed source models such as GPT-4o. We demonstrate that augmenting real data with synthetic data from Bi-Gen enables a $50\%$ reduction in the amount of real data required to learn a classifier for a downstream task.

## 2 RELATED WORK

**LMMs in driving**: Recent advancements in multi-modal integration in LLMs, such as Instruct-BLIP (Li et al., 2023) and LLaVA (Liu et al., 2024), have demonstrated significant success in tasks involving both visual and textual data, showcasing their potential for interpreting and generating meaningful content across modalities. Aligning text with a single additional modality often involves training a projection layer (Li et al., 2024a; Luo et al., 2023) to map features from the new modality into the language space. Most prominent efforts to integrate multiple modalities within unified LLMs primarily target visual-language tasks, such as visual question answering, object detection and image-text similarity (Alayrac et al., 2022; Girdhar et al., 2023; Peng et al., 2023; Wang et al., 2023; Ye et al., 2024).

Prior work as begun to extend the application of LMMs to driving tasks, particularly for AVs. These efforts build on the success of LMMs in visual-language tasks to improve the understanding of driving scenarios captured by onboard cameras and then generate the control signals (Xu et al., 2024; Sima et al., 2023; Wu et al., 2023). These control signals are treated as the same modality as the text domain, without requiring decoder transformation (i.e., controls are specified in language). However, this approach may face challenges when applied to long-horizon trajectory prediction due to context window or memory constraints. LMMs in prior AV work also focus on single-turn interactions, neglecting longer form conversations or rollouts. Moreover, the use of LMMs for learning human driving behaviors remains less studied, and the scarcity of large-scale paired trajectory-language datasets poses a significant challenge to advancing LMMs in this domain. The work presented in this paper, to the best of our knowledge, is the first attempt to integrate multi-modal inputs into LLMs to generate a diverse range of multi-modal outputs for the *multi-turn* inference in the human driving domain. Our approach enables multi-turn question-answering tasks that seamlessly alternate between trajectory description and generation, enabling annotation or generation of trajectory-language paired data in low-data regimes.

**Trajectory-Language Interactions**: Recent works focus on leveraging LLMs in trajectory generation, enabling AVs to make informed, contextually aware decisions in real-time, and guiding low-level motion control to enhance both safety and operational efficiency (Seff et al., 2023; Nguyen et al., 2024; Xu et al., 2024; Chen et al., 2023b). However, such approaches may introduce modeling biases that might fail to capture rare events (e.g., generating unsafe trajectories). Prior works (Nguyen et al., 2024; Kwon et al., 2023; Hu & Sadigh, 2023; Zhang et al., 2024c) addressed this by incorporating a reinforcement learning agent to assess behavioral alignment with different trajectories by the finite state rewards. These works heavily rely on retrieval-augmented generation to provide sufficient context about trajectories. While they perform well in static or deterministic settings, but they have struggled to generalize to interactive environments characterized by high stochasticity and unpredictability. Our work focuses on understanding the diversity and stochastic nature of human driving behaviors, and introduces a multi-modal pipeline based on LMM tuning. This pipeline

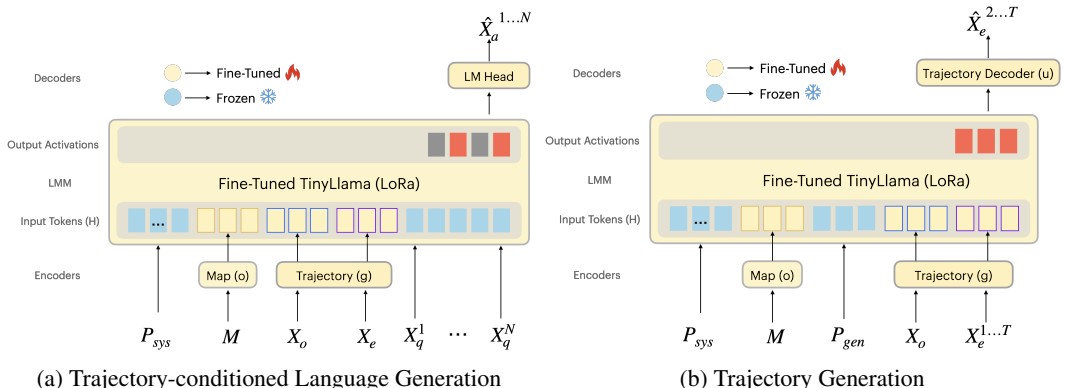

(a) Trajectory-conditioned Language Generation    (b) Trajectory Generation

Figure 2: This figure depicts the two task-setups used to train Bi-Gen. (a) For the trajectory-conditioned language generation task, Bi-Gen is trained to answer a set of questions given the map, $\mathbf{M}$, ego-trajectory, $\mathbf{X}_e$, and opponent-trajectory, $\mathbf{X}_o$. (b) For the trajectory generation task, Bi-Gen is trained to auto-regressively predict the output trajectory, $\hat{\mathbf{X}}_e$ based on an generation prompt, $\mathbf{P}_{gen}$, the map, $\mathbf{M}$, and the trajectory of the opponent, $\mathbf{X}_o$.

enables bidirectional generative modeling, achieved through both language-conditioned trajectory generation and trajectory-conditioned language generation.

## 3 LEARNING A MULTI-MODAL MODEL: BI-GEN

We extend LLaVa (Liu et al., 2024) to a multi-modal architecture as shown in Fig. 2. For both the trajectory and language generation tasks, the model consumes an ego-agent driving trajectory $\mathbf{X}_e^{1:T}$, a task-specific prompt $\mathbf{P}$, and any available driving environment information. We differentiate between global, static information (e.g., map, road conditions) and local, dynamic information (e.g., movement of surrounding objects, vehicles, or pedestrians) in the driving environment. For a typical annotation task given to the model (i.e., to describe the ego vehicle's behavior), the multi-modal input consists of a system prompt, $\mathbf{P}_{sys}$, static map information $\mathbf{M}$, a dynamic opponent vehicle $\mathbf{X}_o^{1;T}$ and an ego-centric trajectory $\mathbf{X}_e^{1:T}$.

To process the driving data, our model employs a trajectory encoder, $g(\cdot)$, that is responsible for embedding both opponent and ego driving data into the LLM's latent space. Specifically, for an input sequence of trajectory states, $\mathbf{X}^{1:T}$, we embed $\mathbf{X}$ into the embedding space of the language model, creating a sequence of $T$ trajectory tokens, $H_\tau$. This process is repeated for each vehicle in the scene, translating from trajectory features into trajectory tokens that the LLM can interpret.

Similarly, a map encoder, $o(\cdot)$, embeds relevant map data for each sequence. As with trajectory data, we pass a sequence of points of map data in, $\mathbf{M}_{1:K}$, and the resulting map embedding is a sequence of $K$ map tokens, $H_\phi$. Prior to embedding, the map data and all driving trajectories (ego and any opponents) are normalized into the same coordinate frame.

We built our model on top of TinyLlama (Zhang et al., 2024b), a lightweight 1B model, for easy deployment and fast inference. The static map and trajectory encoders, $o$ and $g$, are also lightweight networks, constructed as 2-layer multi-layer perceptrons (MLPs) with residual connections. Finally, our model includes a trajectory decoder, $u(\cdot)$, which is an MLP composed of two linear layers with ReLU activation. This trajectory decoder is designed to project from the LLM's hidden dimensions down to the original trajectory dimension.

### 3.1 TRAJECTORY DESCRIPTION

The trajectory description task is formulated as a multi-turn question-answering task. The model is given a system prompt, which includes a short task description and all relevant scene information (all driving trajectories and map information). Finally, a language sequence, $X_L$, is passed to the model as a series of $N$ questions and answers, $(X_q^1, X_a^1, \ldots X_q^N, X_a^N)$, where $X_q^1$ and $X_a^1$ are the

| Trajectory Description | Trajectory Generation |
|---|---|
| *System Prompt:* $\boldsymbol{P_{sys}}$

The track is: **\<M\>** The trajectory of the opponent is: $< \mathbf{X}_O^{1:T} >$ The ego trajectory is: $< \mathbf{X}_e^{1:T} >$
*User:* $\boldsymbol{X_q^1}$
*Assistant:* $\boldsymbol{\widehat{X}_a^1}$
*User:* $\boldsymbol{X_q^2}$
*Assistant:* $\boldsymbol{\widehat{X}_a^2}$ | *System Prompt:* $\boldsymbol{P_{sys}}$

The track is as follows: **\<M\>** The description is: $< \boldsymbol{P_{gen}} >$ The trajectory of the opponent is: $< \mathbf{X}_O^{1:T} >$ The generated trajectory is: $< \mathbf{X}_e^1 >$
*Assistant:* $\boldsymbol{\widehat{X}_e^{2:T}}$ |

Figure 3: For trajectory description (left), the model is trained to interpret the map (orange), the opponent's trajectory (blue), and the ego trajectory (purple) to answer questions accurately. For trajectory generation (right), the model processes the same map and opponent information, along with a description prompt (green), to produce the user-requested trajectory. $\mathbf{X}_a^n$ and $\mathbf{X}_e^{2:T}$ are used to compute the losses used to train Bi-Gen.

first question and answer respectively. The model is then tasked with autoregressively predicting all $X_a$ utterances. We mask out all questions, $X_q$, from the target sequence to prevent the model from learning to play both sides of the conversation (i.e., learning to ask questions and answer them).

Fig. 2a provides a visual overview of this task. The complete input sequence to the model, $\mathbf{H}$, is given as the concatenation of LLM embeddings for the text in the system prompt, $\mathbf{H}_P$, embedded map data, $\mathbf{H}_\phi$, embedded trajectory data for the opponent and ego agents, $\mathbf{H}_{o\tau}$ and $\mathbf{H}_{e\tau}$, and the LLM embeddings for the question-answer sequence, $\mathbf{H}_{QA}$. This complete input sequence is then passed through an LLM, and the LLM is tasked with predicting the answer tokens in the sequence. We apply a LoRA (Hu et al., 2022) to tune the LLM to the task of trajectory description. Gradients are computed as the language modeling loss between the predicted answers and the ground-truth targets, and are applied to the map and trajectory encoders, $o$ and $g$, and to the LLM via the LoRA.

## 3.2 TRAJECTORY GENERATION

To learn to generate new data in the driving domain, we formulate the trajectory generation task as shown in Fig. 2b. Similar to the trajectory description task, the input is a sequence consisting of a system prompt, static map information, dynamic opponent vehicle information, and an ego trajectory. Unlike the trajectory description task, here we do not use any question-answering text input, and the loss function is not a language modeling objective. Instead, we use the generation prompt $\mathbf{P}_{gen}$ as the input and train a decoder, $u(\cdot)$ that learns to map from an LLM hidden state back into trajectory features. The training objective is an autoregressive mean-squared error (MSE) loss between the predicted trajectory output, $\hat{\mathbf{X}}_e^{2:T}$ and the actual trajectory input, $\mathbf{X}_e^{2:T}$.

## 3.3 TRAINING PROCEDURE

The two tasks of trajectory description and trajectory generation can be trained independently to achieve one-directional data generation (i.e., learning to annotate trajectories with language or learning to generate new trajectories from language prompts). However, training the two tasks together enables the model to both see a greater diversity of data as well as learn to leverage the shared structure of the two tasks to learn a more robust model of the relationship between language and driving data. We therefore train the two tasks jointly with a single model (Fig. 2).

As described in Sec. 3.1 & 3.2, the input token sequence includes static map information and opponent and ego trajectory data. The static map information includes inner and outer edges of the track, centered around the ego agent's location. Trajectory data for both agents includes three-dimensional position and velocity and a quaternion for orientation. As mentioned above, trajectory and map data are all normalized to the same coordinate frame before being passed to the model, and are further normalized to have a zero-mean and unit variance for numerical stability. While future work may consider more sophisticated feature normalization or unification strategies to further improve per-

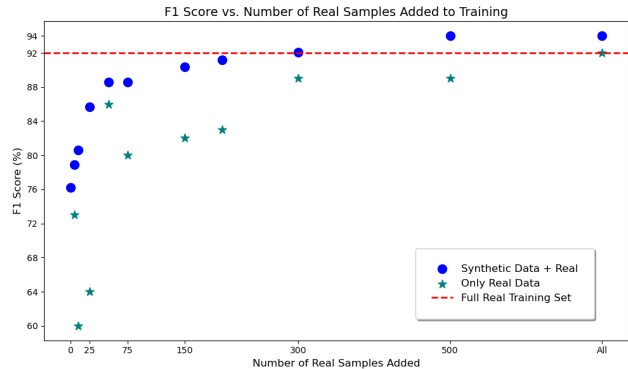

Figure 4: F1 for an overtake prediction task when training a model on synthetic data + real data vs. training only on real data, and testing on unseen, real data. We show that training on synthetic data from Bi-Gen reduces the training data requirements for learning to classify overtakes. Adding synthetic data *always* improves performance on the task compared to only using with real data.

formance, such as cross-attention between map and trajectory data (Kuo et al., 2022), we found a simple projection and self-attention strategy to be sufficient in this work.

For the trajectory description task, we randomly sample a set of six questions for each multi-turn conversation (Table 1 in Appendix A), and we randomly sample one prompt for each trajectory generation sample (Table 3 in Appendix B).

We present a visual example of a multi-modal input sequence for the trajectory description task in Fig. 3 (left). The model is trained to interpret static map information (orange), dynamic opponent information (blue), and the ego-agent trajectory (purple) to answer questions accurately. For the trajectory generation task (Fig 3, right), the model processes the same static and dynamic information, along with a generation prompt (green), to produce the user-requested trajectory. The predicted tokens (red) are used to compute the auto-regressive losses to learn the model.

The joint training objective for our model is a sum of two objectives. The first is a language modeling loss, $L$, between the predicted answers, $\hat{\mathbf{X}}_a$, and ground-truth answers, $\mathbf{X}_a$, for the trajectory description task. The second is a mean squared error loss, $MSE$, between the reconstructed trajectory features, $\hat{\mathbf{X}}_{2:T}$ and ground-truth trajectory features $\mathbf{X}_{2:T}$ for the trajectory generation task.

$$loss = w_1 L(\hat{\mathbf{X}}_a, \mathbf{X}_a) + w_2 MSE(\hat{\mathbf{X}}_e^{2:T}, \mathbf{X}_e^{2:T})$$

In this equation, $w_1$ and $w_2$ are task-specific weights, though we found empirically that the simple setting of $w_1 = 1, w_2 = 1$ worked well.

Bi-Gen employs an end-to-end training approach, consisting of two main steps: (1) mapping the additional modalities into the LLM's feature space by learning robust encoders and decoders, and (2) tuning the entire network to extend the LLM's implicit world-knowledge and pattern-recognition to the new modalities and tasks. The entire process can be understood as training a combination of compatible tokenizers to align different modalities with the LLM's language space, and fine-tuning the LLM to maximize its ability to exploit these new data sources. This combination of learning processes enables bi-directional mapping and modeling between language and trajectory data.

## 4 EXPERIMENTS AND RESULTS

We evaluate our model's performance in a low-data human driving domain, using training and testing data sourced from a high-performance multi-car racing environment from prior work (Anonymous, 2024). More details on data collection and curation in the racing domain can be found in the Appendix A. We evaluate Bi-Gen as both an annotator and a synthetic data generator.

To test Bi-Gen as an annotator, we use a trained model to annotate unseen, unlabeled racing trajectories. We formulate this task as a 9-turn conversation, with randomly shuffled questions drawn

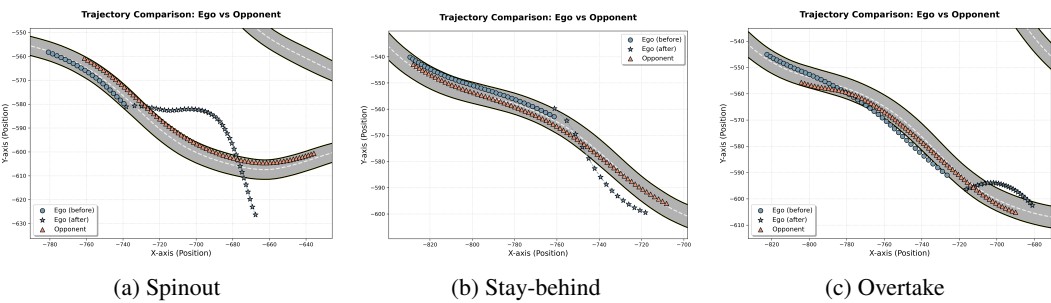

(a) Spinout  (b) Stay-behind  (c) Overtake

Figure 5: We show that Bi-Gen can complete a partial trajectory, conditioned on a language prompt. Here, we prompt Bi-Gen with a complete opponent trajectory and partial context for an ego trajectory, and then provide a language description of the desired outcome. Qualitatively, we see that generated synthetic data reflects the provided language prompt.

from the same distribution as the training data (note that the trajectories are entirely unseen for the model). We then score the model's ability to generate the appropriate answer for each question using F1 with all 19 classes of possible answers (including a class for "nothing" if the model generates unrelated text).

To test Bi-Gen as a synthetic data generator, we task the model with generating both seen and unseen trajectories to create an entirely synthetic dataset of racing trajectories. We then use this dataset to train a binary classifier on an overtake prediction task (i.e., does the ego-agent overtake its opponent in this clip?). We evaluate this classifier on held-out *real* data, thereby measuring how accurately the *synthetic* data distribution approximates the real data distribution.

Finally, we qualitatively test Bi-Gen as a full, end-to-end system as annotator and synthetic data generator. We first task the model with answering questions about an unseen trajectory, and then ask the model to convert the given trajectory into a new trajectory that looks different (e.g., turn a safe trajectory into a spinout, or turn a stay-behind into an overtake).

### 4.1 ANNOTATOR EXPERIMENT

We first evaluate Bi-Gen as an automated labeler or annotator for entirely unseen and unlabeled trajectory data. In this setting, we formulate the task as a multi-turn conversation, reflecting a possible deployment of our lightweight model to a real-time annotation platform. For each new sample consisting of a system prompt, map, opponent trajectory, and ego-trajectory, we ask questions one-at-a-time, allowing the model to generate a short response to each question before moving on to the next question. Each sample is followed by nine total question-answer pairs.

Note that this task is more challenging than a conventional annotation task because of the sequential, multi-turn structure of the evaluation. If the model generates an unrelated response early, or begins to wander out-of-distribution, the entire conversation can collapse. Therefore, the model must remain accurate for all nine turns to maximize its score.

We compare Bi-Gen to GPT-4o (Achiam et al., 2023) as a baseline for a closed-source, expensive, large baseline model. GPT-4o is given sub-sampled trajectory data in the form of a JSON, and is asked to answer all questions simultaneously. For both our model and GPT-4o, generated answers are binned into target topics using GPT-4o as a judge. We provide further details on this process in Appendix C. After this binning, the F1 score for our model is 43.9%, comparable with the GPT-4o's score of 43.8%. Despite using a fraction of parameter count of a state-of-the-art LLM, Bi-Gen is able to achieve comparable auto-annotation accuracy, and is even able to do so in a multi-turn conversational format.

### 4.2 SYNTHETIC DATA GENERATION

We next evaluate Bi-Gen as a synthetic data generator to produce new ego-centric trajectories that approximate real data. To quantitatively evaluate the fidelity of these new trajectories, we set up a binary classification task in which the objective is to predict whether or not the ego agent has

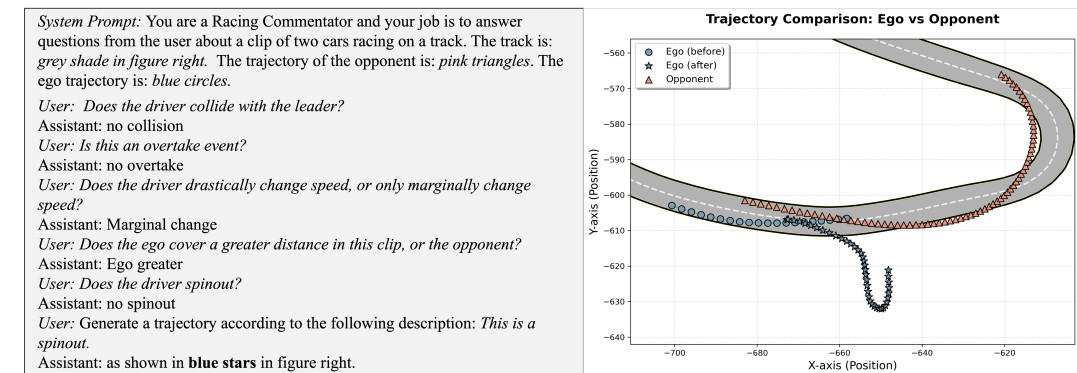

Figure 6: This end-to-end pipeline example shows that Bi-Gen can comprehend the input trajectories and track to accurately answer user questions regarding the ego-agent's trajectory (e.g., overtakes, spinouts, collisions, etc.). The model can then generate a new ego trajectory to convert the given no-spinout trajectory into a spinout trajectory based on the user's query, satisfying the user's request.

overtaken its opponent. If the synthetic data closely approximates the real data distribution, then a model that is trained on synthetic data should perform well on real data.

We first generate a synthetic dataset using snippets of trajectories from the training set as context, and tasking the model with generating completions that conform to either an "overtake" or a "stay-behind" prompt. We then train a small long short-term memory (LSTM) model (Hochreiter & Schmidhuber, 1997) to perform the binary "overtake" or "stay-behind" prediction. We train this model on different mixtures of data, including entirely synthetic, entirely real, and synthetic with small amounts of randomly sampled real data mixed in. For each data mixture, the model is evaluated on entirely unseen real data.

In Fig. 4, we compare the performance of this classifier trained different data mixtures. We see that purely synthetic data achieves quite strong performance, though it lags behind training on the full real training set. However, by adding small amounts of real data to the synthetic dataset, we are able to quickly match and even exceed the performance of a real-data-only classifier.

When training with similarly small amounts of only real data, we see that the downstream classifier always lags behind a model trained on the mix of synthetic and real data, highlighting the performance boost that comes from using synthetic data from Bi-Gen. This result highlights the strength of Bi-Gen, as it enables us to cheaply augment and extend small, real datasets with larger amounts of synthetic data that can lead to performance gains over using a smaller, entirely real dataset.

### 4.3 MULTI-MODAL GENERATION

Here, we present a qualitative demonstration of Bi-Gen as a trajectory generator and when deployed to a full, end-to-end data generation setting. In this full end-to-end setting, we first test the model's ability to handle new phrasings of questions that it has seen before, and then we ask the model to generate a different completion to a given input trajectory. Note that this task is never encountered during training, and this combination of tasks in a single interaction is also never encountered during training. While the model is trained on both tasks in the same batch, there are no training examples of two tasks in one conversation.

First, we present qualitative examples of trajectory completions from our model when applied to new, unseen input data (i.e., unseen opponent and ego trajectories). We present the model with a complete trajectory from each agent, and then task the model with generating a new completion to the ego trajectory, conditioned on a language prompt (e.g., "This driver overtakes the opponent:"). Examples of this process are shown in Fig. 5, where we show our model generating novel completions as either a spinout, a stay-behind, or an overtake. Note that the latter two examples require reasoning about both the ego and opponent trajectories, forcing the model to generate data that ac-

curately satisfies the requested relationship between the two agents (i.e., stay behind the opponent or overtake the opponent).

Finally, we demonstrate Bi-Gen's ability to handle multi-turn question-answering and trajectory generation in Fig. 6. This example shows that Bi-Gen can comprehend the input trajectories and track to accurately answer user questions regarding the ego-agent's trajectory (e.g., overtakes, spinouts, collisions, etc.). Even without exactly matching phrasings that the model has been trained on, the model is able to generate completions that accurately reflect the given trajectory data. Finally, the model can further generate a *new* ego trajectory to convert the given "no-spinout" trajectory into a "spinout" trajectory based on the user's query, satisfying the user's request. An additional example is provided in Appendix E.

## 5 CONCLUSION

In this work, we presented Bi-Gen, a bi-directional large multi-modal model designed to bridge the gap between trajectory description and generation. By leveraging the strength of LLMs, we demonstrated that Bi-Gen can serve as an automated annotator or synthetic data generator, providing rich augmentations to existing data that prove particularly valuable in low data regimes, such as multi-car racing.

The bi-directionality of Bi-Gen marks a significant advancement over prior works, which focus on a single aspect of the trajectory modeling problem. By leveraging the joint structure and complementary data for trajectory description and generation, Bi-Gen is able to learn a richer understanding of trajectory data, leading to an enhanced ability to bolster existing datasets.

Looking forward, Bi-Gen is a step towards further advancements in multi-modal modeling in embodied domains. As a framework for integrating multiple modalities for real-time data synthesis and understanding, Bi-Gen could be applied to driving, robotics, or other digital interaction domains. Future work may seek to further extend Bi-Gen with additional modalities, such as video or other real-time sensors, expanding the capabilities of Bi-Gen to more complex data synthesis and modeling problems.

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

## A    HUMAN DRIVING DATA COLLECTION IN MULTI-CAR RACING DOMAIN

While the data collection and dataset are not contributions of this work, we briefly discuss them here for completeness and clarity in the submission. The dataset was captured during a user study, which was designed to gather human driving behavior data in the racing domain we use in the paper. The purpose of the study was to gather qualitative and statistical data on individuals' behavior and objectives in a racing context, and to use that to inform what criteria are important for building models of human objectives. We recruited 48 participants to drive a simulator with the hairpin and straightaway segments of the two-mile racing track, the same domains for the computational results in this paper. The scenarios were chosen so as to present overtake opportunities in portions of the track of varying levels of difficulty, while keeping the overall task short enough to ensure there is a rich interaction between the ego and opponent. Participants completed a series of warm-up trials in each domain, with three trials devoted to the straightaway segment and eight trials in the hairpin segment, each featuring different opponents of varying difficulty (fixed trajectories) to race against. Again, these were the same trajectories used in our domains. At the conclusion of each trial, participants answered the question: "Did you attempt to pass the other vehicle?" on an iPad. We also gathered, from trajectory data, whether or not the participant actually completed an overtake without collisions or spin-outs. 877 trajectories are collected. We then further manually label the data to address nine specific questions, as outlined in the Table 1 to construct our question-answering set.

Table 1: Questions for Labelling

| No. | Questions |
|---|---|
| 1 | Does the driver attempt to overtake? |
| 2 | Does the driver cheat across the track? |
| 3 | Does the driver collide with the leader? |
| 4 | Is this an overtaking event or a stay-behind event? |
| 5 | Is there any spinout? |
| 6 | Is the driver going faster in the first half or second half of the trajectory? |
| 7 | Is the driver closer to the opponent in the first half or second half? |
| 8 | Are there any drastic changes in the driver's speed? |
| 9 | Does the driver cover a greater distance over the course of the trajectory? |

We present two empirical examples from the collected human racing data, supported by human-labeled ground truth multi-turn QAs in Table 1: Fig. 7 illustrates the ego vehicle staying behind the opponent, while Fig. 8 depicts an overtaking maneuver.

## B    TRAINING DETAILS

The model's training hyperparameters are listed in Table 2.

Table 2: Training Hyper-parameters

| Hyper-parameter | Values |
|---|---|
| Learning rate | $5e^{-5}$ |
| Trajectory Encoder Hidden Dim | 128 |
| Map Encoder Hidden Dim | 128 |
| Trajectory Decoder Hidden Dim | 128 |
| Batch Size | 16 |
| Training Epochs | 20 |
| LoRA Layers | $Q, K, V, O$, up-projection, down-projection |
| LoRA Rank | 16 |
| LoRA Alpha | 16 |
| LoRA Dropout | 0.1 |

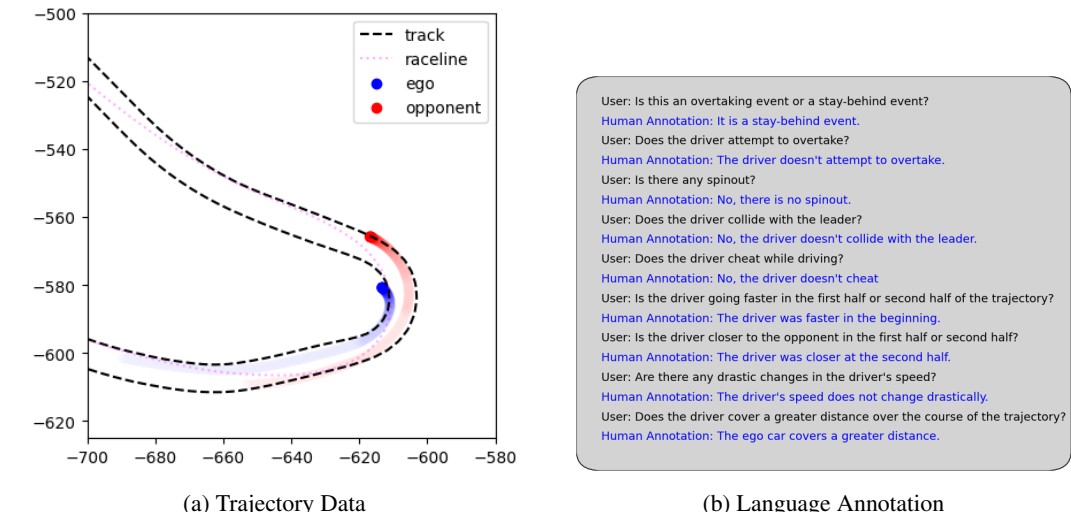

(a) Trajectory Data      (b) Language Annotation

Figure 7: Here, we show an example of training data used for Bi-Gen. Trajectories are collected from a small in-person user study, in which participants are instructed to attempt to overtake an automated racing opponent. After collecting hundreds of small trajectory clips (approximately 800 total clips), a human annotator reviewed each clip to label specific events such as overtakes, spinouts, collisions, etc., while automated heuristics created labels for data about the trajectory statistics (speeds, distances, etc.).

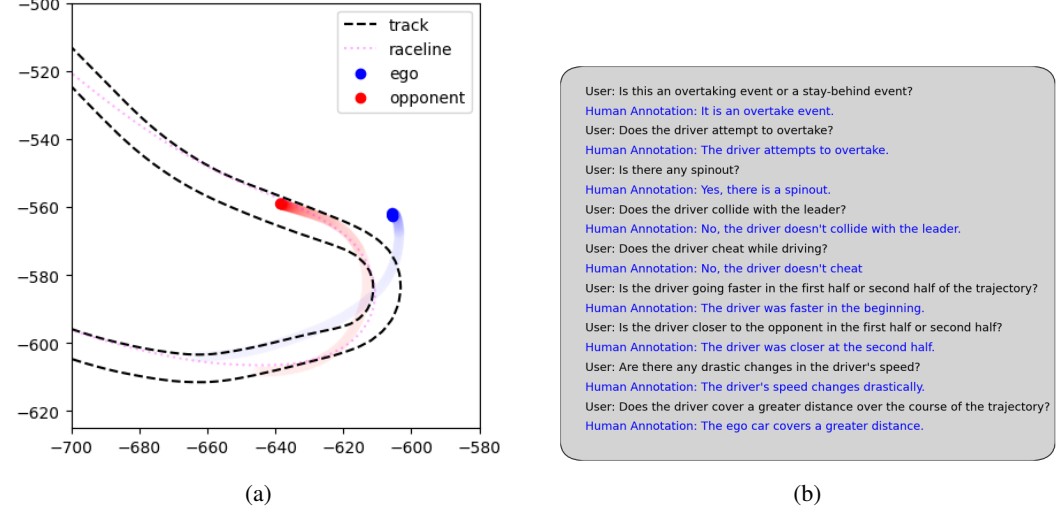

(a)      (b)

Figure 8: Here, we show another example of training data used for Bi-Gen, this time for an overtake that leads into a spinout. This example shows the complexity of the data, as a model or annotator must watch the data unfold in realtime to catch overtakes. Simply viewing the end product might show something completely different (such as this driver, who spun out after briefly overtaking the opponent).

The descriptive prompts for trajectory generation are summarized as Table 3.

## C  F1 SCORE CALCULATION USING GPT AS A JUDGE

The detailed pipeline for using GPT as a judge to evaluate Bi-Gen's descriptive capabilities is illustrated in Fig. 9. The process for obtaining predicted and ground truth label classes for F1 score calculations involves two steps:

Table 3: Descriptive Prompts for Trajectory Generation

| No. | Descriptions |
|---|---|
| 1 | The ego car was faster in the second half. |
| 2 | There is no spinout. |
| 3 | The driver cheats. |
| 4 | The driver was closer in the second half. |
| 5 | The driver was faster in the beginning. |
| 6 | The driver doesn't collide with the leader. |
| 7 | It is a stay-behind event. |
| 8 | The ego car covers a greater distance. |
| 9 | The driver was closer in the first half. |
| 10 | The driver collides with the leader. |
| 11 | The driver's speed does not change drastically. |
| 12 | The opponent covers a greater distance. |
| 13 | The driver doesn't cheat. |
| 14 | The driver's speed changes drastically. |
| 15 | It is an overtake event. |
| 16 | The driver attempts to overtake. |
| 17 | The driver doesn't attempt to overtake. |
| 18 | There is a spinout. |

**Step 1: Constructing a Topic Pool**: Ground truth answers are manually labeled and categorized into several key topics, with each topic assigned a corresponding class number. For example, a ground truth answer such as "It is an overtake event" is classified under the topic "overtake," and its label is marked as "1."

**Step 2: Topic Selection using GPT**: The predicted answers are fed into GPT, and it is prompted to select the most appropriate topic from the constructed topic pool. Then topic labels are assigned to the predicted answers.

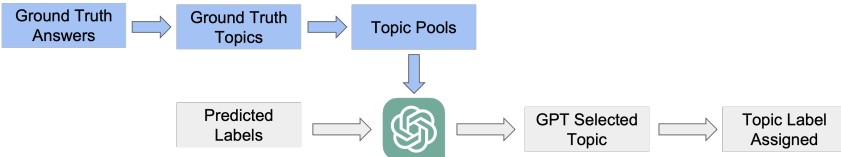

Figure 9: We use GPT-4o as an automated labeler to assign class labels to arbitrarily generated text from our model, and from GPT-4o as a baseline method. Unconstrained generations are passed to GPT-4o along with a list of possible class labels (topic pools), and GPT-4o must return an appropriate topic, if one can be found.

Based on the ground truth labels assigned to the human driving data, as detailed in Appendix A, the constructed topic pool of 18 topics along with the percentage distribution is illustrated in Fig. 10.

## D GPT 4O FOR MULTI-TURN QA

We prompt GTP-4o to conduct the multi-turn QA task performed by Bi-Gen. Leveraging GPT-4o enables us to compare the performance of Bi-Gen with a large-scale, multimodal language model.

We begin by downsampling the original trajectory, selecting one out of every 25 points. This downsampled trajectory is then converted to a JSON formatted dictionary and given GPT-4o to facilitate multi-turn QA about the given trajectory. The full prompt utilized is provided in Fig. 11. Unlike Bi-Gen, GPT-4o was provided the list of topics to chose from while answering each question. We optionally included an image plotting the trajectory on a 2D graph, and zero-shot chain-of-thought (CoT) reasoning in the prompt, however, we found that the best performing version of GPT-4o did not utilize images or CoT.

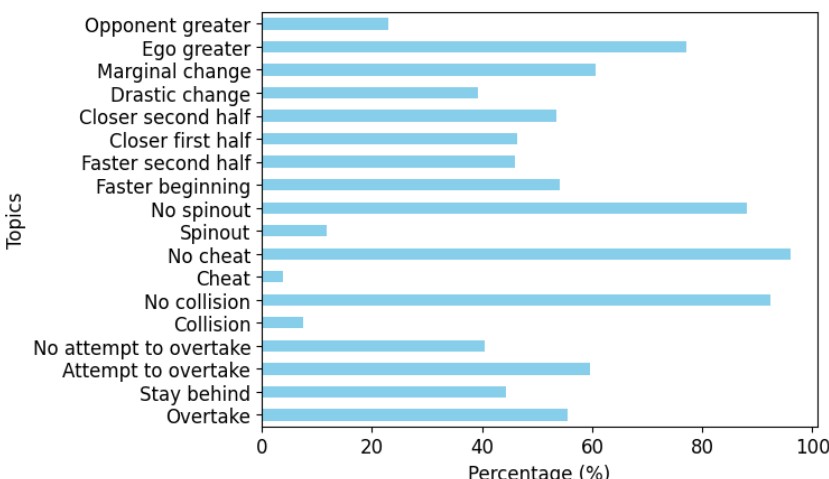

Figure 10: Here we show a breakdown of the percent of each label or topic in the dataset. While some classes are extremely imbalanced (such as collisions or cheats), others are nearly 50-50 (such as overtakes or speed differences).

---

**Prompt for answering question based on a trajectory.**

Your job is to answer a given set of questions about a trajectory of two race-cars on the track.

The trajectory will be provided to you in a json format. The json dictionary is indexed by timesteps in the trajectory. At each timestep you will be provided the position of the "ego" car and the "ado" car which is the adversary. You will be given the positions and velocities with regards to the x,y,z axes for both cars. You are required to answer questions from the perspective of the "ego" car.

Your answer to any question must be selected from the topic pool in Appendix C:

Please utilize the json trajectory, and the image plotting the trajectory, to answer the following questions.:
**Insert Questions**

Your answer should be in the following format:
# # #
**Answer1**: ...
**Answer2**: ...
...
# # #

---

Figure 11: Full prompts utilized in GPT4o for multi-ture QA

## E   MULTI-MODAL GENERATION

Fig. 12 presents an additional example demonstrating Bi-Gen's multi-turn inference capability in trajectory description and generation. This figure shows that Bi-Gen can comprehend the input trajectories and track to accurately answer user questions regarding the ego-agent's trajectory (e.g., spinouts, speed features, etc.). The model can then generate a new ego trajectory to convert the given stay-behind trajectory into an overtake trajectory based on the user's query, satisfying the user's request. Note that the phrasing of the questions in the multi-turn conversation is different from the questions in the dataset (Table 1), though the model is able to effectively generalize its

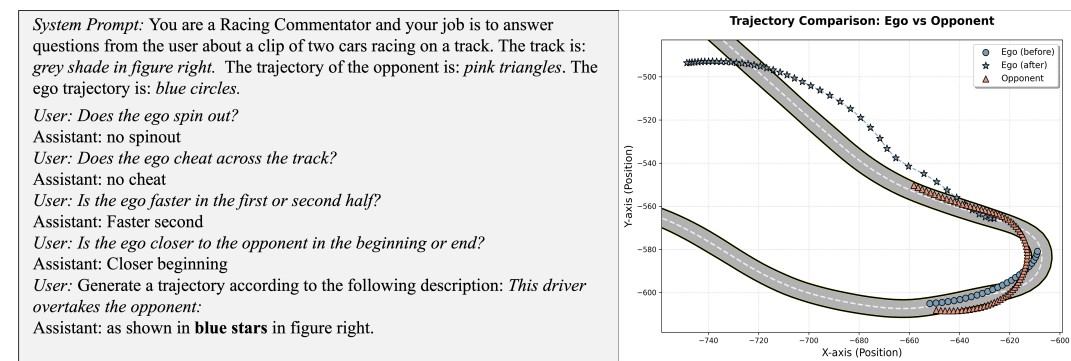

*System Prompt:* You are a Racing Commentator and your job is to answer questions from the user about a clip of two cars racing on a track. The track is: *grey shade in figure right.* The trajectory of the opponent is: *pink triangles*. The ego trajectory is: *blue circles.*

*User: Does the ego spin out?*
Assistant: no spinout
*User: Does the ego cheat across the track?*
Assistant: no cheat
*User: Is the ego faster in the first or second half?*
Assistant: Faster second
*User: Is the ego closer to the opponent in the beginning or end?*
Assistant: Closer beginning
*User:* Generate a trajectory according to the following description: *This driver overtakes the opponent:*
Assistant: as shown in **blue stars** in figure right.

Figure 12: This end-to-end pipeline example shows that Bi-Gen can comprehend the input trajectories and track to accurately answer user questions regarding the ego-agent's trajectory (e.g., spinouts, speed features, etc.). The model can then generate a new ego trajectory to convert the given stay-behind trajectory into an overtake trajectory based on the user's query, satisfying the user's request.

learned racing-knowledge with the help of the inherent commonsense reasoning of the pre-trained TinyLlama LLM.

When generating the multi-turn conversations for Fig. 6 & 12, we manually enter text queries to the model, and allow the model to generate for a fixed number of tokens. After 5 turns, we inject a small prompt to the model to more closely reflect the language prompting in the training data, and then we manually enter a description of the desired trajectory generation.