# OpenReview forum: "Learning a Bi-directional Driving Data Generator via Large Multi-modal Model Tuning"
_ICLR.cc/2025/Conference — ICLR 2025 Conference Withdrawn Submission_

### Official Review · Reviewer_n3mP · 2024-10-31

**Soundness:** 2
**Presentation:** 1
**Contribution:** 2
**Rating:** 3
**Confidence:** 4

**Summary:**

The manuscript presents an exploration of finetuning a TinyLLaMa for generating multi-modal human driving data to benefit the community. While the proposed solutions sound acceptable, the motivation, experiment results, readability, and method description should all be improved. So far the weakness outweighs the strengths.

**Strengths:**

An acceptable exploration in using LLMs to benefit understanding driving data.

**Weaknesses:**

1. Unmatched motivation and proposed solutions. To my understanding, the stated motivation in the abstract is the lack of multi-modal data and the high cost of obtaining labeled data for training, but the proposed solution is a model that can match the performance of LLMs but can be adopted in a resource-constrained setting. There is a gap between them. In the context of generating multi-modal data, why do we need a resource-constrained model? Do the authors try to generate data during driving, or deploy the generation model in the vehicle? If not, why is there a need to design such a model?

2. Insufficient contribution. As the authors said between lines 88 and 97, the existing LLMs cannot fully comprehend the complicated multi-modal connections between trajectories and languages due to the lack of readily accessible world-knowledge. Hence, we would expect that with the proposed solutions, the generation performance should at least outperform the existing LLMs, despite the model size. But so far, the annotation performance is only compatible, and hence the proposed solution is not as effective as the authors claimed.

3. Poor readability in terms of images and texts. The images are not aligned with the text around it. For example, Fig. 3 is too far away from the text describing it. Fig. 4 is 2-pages away from the corresponding text. Readers may get confused by the images and find it difficult to find the text and hence fail to follow.

4. Unclear method description. Maybe I miss something. In the trajectory generation part, the loss is the auto-regressive loss between the generated trajectories and the actual ones. If the authors aim to fine-tune the model based on this task, then we are assuming that the model is the core component controlling how the trajectories are generated. But as we include human languages or prompts here, is it possible that the inputs are affecting the generating performance? Do we consider any loss in terms of the discrepancy between the generated trajectories and what the prompts ask for?

**Questions:**

See weakness.

---

### Official Review · Reviewer_6U3U · 2024-10-31

**Soundness:** 1
**Presentation:** 1
**Contribution:** 1
**Rating:** 3
**Confidence:** 3

**Summary:**

The paper presents Bi-Gen, a large multi-modal model designed to generate and annotate human driving data, particularly in complex racing environments with limited training data. It effectively handles both trajectory description and generation, demonstrating strong performance in comprehending driving behaviors. The study highlights the model's ability to produce realistic and varied driving scenarios, positioning it as a competitive alternative to larger models like GPT-4o.

**Strengths:**

The paper tackles the issue about interpreting and annotating unlabeled driving trajectories in the low-data domain of high-performance
multi-car racing.

**Weaknesses:**

1. The model's performance in more diverse and complex driving scenarios beyond the tested environments may require further exploration and validation.
2. The evaluations appear to be inadequately conducted. The authors assert the existence of 19 potential answer classes; however, they report only the quantitative results from the overtaking prediction task. Furthermore, the zero-shot testing with GPT-4o is the sole baseline selected for comparison. There are also no numbers supporting the claim that training trajectory description and trajectory generation at the same time would be a more favorable approach.

**Questions:**

1. Both the training and test data are collected within a single racing track by driving in simulators (as indicated in Appendix.A). The number of trajectories collected are limited as well. How do you ensure that your model does not overfit to this narrowly defined domain?

---

### Official Review · Reviewer_PZJ4 · 2024-11-03

**Soundness:** 2
**Presentation:** 3
**Contribution:** 2
**Rating:** 3
**Confidence:** 5

**Summary:**

This paper identifies that learning driving behaviors requires a lot of data with carefully labeled events, causes, and consequences. However, such data may be more difficult to obtain in rare driving domains, such as in high-performance multi-car racing. Therefore, this paper proposes Bi-Gen, which is a bi-directional multi-modal model that connects a trained encoder-decoder architecture with a pre-trained LLM, enabling both auto-annotation and generation of human driving behaviors. The experimental results show that Bi-Gen matches the performance of much larger models like GPT-4o in annotating driving data. Additionally, Bi-Gen generates diverse, human-like driving behaviors, offering a valuable tool for synthetic data generation in resource-constrained settings.

**Strengths:**

1.	The idea of using LLM to generate scenarios is an interesting and promising topic. Since LLMs have limitations on processing other modalities, finetuning LLMs is also a promising way to direct them to the generation task.
2.	The paper is generally well-written and well-organized. Figure 1 clearly shows the training and generation processes of the proposed method. Figure 2 also describes the two generation tasks with model details.

**Weaknesses:**

1.	The examples shown in Figure 5 are quite confusing. First, the generated trajectories violate the vehicle dynamics and are out of road in most times. The first point of the generated trajectory in the middle figure is behind the last point of the history trajectory. The first point of the generated trajectory in the right figure is too far away from the last point of the history trajectory. Second, it is hard to identify Spinout, Stay-behind, and Overtake in these figures. In summary, I think the generated trajectories have low quality and low realism.
2.	I feel the evaluation of this paper is quite limited. It seems that this paper only focuses on high-performance multi-car racing scenarios, as mentioned in the abstract. Even though, I think it is still important to show quantitative results of the average performance of the proposed method. However, the only numerical evaluation now is the overtake classification task shown in Figure 4. I think it is necessary to show the evaluation of realism, diversity, and instruction following. In addition, scenario generation has been a widely investigated area, which means it is easy to find comparable baseline methods, for example, LCTGen [1] and ProSim [2].
3.	There is no evidence to show the benefit of using the generated scenarios for downstream tasks. The only example is the overtake classification task. But I am not sure how large the value is of identifying if a scenario is overtaking or not. I think it is more important to show that the generated scenarios help with the training and testing of autonomous agents in terms of performance and safety.

---
[1] Tan, Shuhan, Boris Ivanovic, Xinshuo Weng, Marco Pavone, and Philipp Kraehenbuehl. "Language conditioned traffic generation." arXiv preprint arXiv:2307.07947 (2023).
[2] Tan, Shuhan, Boris Ivanovic, Yuxiao Chen, Boyi Li, Xinshuo Weng, Yulong Cao, Philipp Krähenbühl, and Marco Pavone. "Promptable Closed-loop Traffic Simulation." arXiv preprint arXiv:2409.05863 (2024).

**Questions:**

1. Did the authors consider different LLM backbones?
2. What are the statistical details of the used dataset? The size, the distribution, and the collection platform?
3. Similar to the second point in the weakness part, how to evaluate the quality of generated scenarios?

---

### Official Review · Reviewer_cJ9k · 2024-11-05

**Soundness:** 1
**Presentation:** 3
**Contribution:** 2
**Rating:** 3
**Confidence:** 3

**Summary:**

This paper introduces Bi-Gen, a bi-directional large multi-modal model that enables both trajectory description (auto-annotation of driving data in language) and trajectory generation. The model leverages a pre-trained LLM and learns to embed multi-modal inputs (map, ego trajectory, opponent trajectory) into a shared latent space. The authors demonstrate Bi-Gen's capabilities on a racing car dataset, showing it can annotate trajectories comparably to GPT-4o and generate synthetic data to augment real datasets for downstream tasks.

**Strengths:**

1. The bi-directional approach allowing both trajectory description and generation within a single end-to-end framework is novel and interesting. Prior work has typically focused on only one direction.
2. The motivation of learning a model that can comprehend and generate multi-modal human driving data, especially in low-data regimes like racing, is sound and the proposed methodology of embedding multi-modal inputs into an LLM's latent space makes intuitive sense.
3. The paper is generally well-written, with a clear explanation of the model architecture, training process, and experimental setup. The figures help illustrate the approach.

**Weaknesses:**

1. While the motivation and methodology are sound, the experimental setup seems too simplistic to fully validate the capabilities of Bi-Gen. The authors mention that there are only 19 possible answers in their question-answering task, which is more akin to a classification problem. This limited setup may not adequately demonstrate the LLM's ability to freely annotate trajectories in an auto-regressive manner.  More open-ended annotation would be valuable.
2. The dataset used for training and evaluation is relatively small, with only 877 trajectories collected. Moreover, the participants were racing against fixed trajectories rather than human players or other agents, which limits the diversity and complexity of the driving behaviors captured. A larger and more varied dataset would provide a more robust evaluation of Bi-Gen.
3. Given the large capacity of LLMs, it is possible that Bi-Gen is overfitting to the training data. The authors do not provide sufficient qualitative results to assess the model's generalization capabilities. It would be beneficial to compare Bi-Gen with a baseline method that uses classification-based annotation and recurrent trajectory generation, rather than solely comparing it with GPT-4o.
4. The binary classifier used to validate the quality of the generated trajectories may not be a strong indicator of performance if the data distribution is too simple. It is unclear whether the higher accuracy achieved by the classifier is due to the quality of the generated trajectories or the simplicity of the data distribution.
5. The paper heavily relies on the supplementary material to provide important details about the methodology and results. Some of this information should be included in the main paper to improve clarity and completeness. Additionally, there is redundant information in the main paper, such as the repeated mention of the model pipeline components (system prompt, map, opponent trajectory, and ego-trajectory).

**Questions:**

1. How would the authors justify the use of a simplistic experimental setup in the question-answering task? Does this truly showcase the LLM's auto-regressive generation capabilities?
2. Have the authors considered collecting a larger and more diverse dataset, possibly including human-human or human-agent interactions, to better capture the complexity of driving behaviors?
3. Can the authors provide more evidance to demonstrate Bi-Gen's generalization capabilities and address concerns about overfitting?
4. Would the authors consider moving some of the important details from the supplementary material to the main paper and removing redundant information to improve clarity and completeness?

---

### Official Review · Reviewer_LN46 · 2024-11-11

**Soundness:** 3
**Presentation:** 3
**Contribution:** 3
**Rating:** 6
**Confidence:** 4

**Summary:**

1. The paper introduces Bi-Gen, a bi-directional multi-modal model for human driving data generation and annotation, particularly aimed at low-data domains like multi-car racing.
2. Bi-Gen combines language-conditioned trajectory generation and trajectory-conditioned language generation, allowing it to serve both as an automated annotator and as a synthetic data generator.
3. The model integrates a language model with lightweight encoders and decoders to map trajectories and static map data into a shared feature space, enabling it to interpret and generate diverse driving behaviors based on limited real data.
4. Experimental results demonstrate Bi-Gen’s ability to match the annotation accuracy of larger models, like GPT-4o, while significantly reducing the data requirements for downstream tasks by generating high-quality synthetic data.

**Strengths:**

1. The model’s ability to handle both trajectory-to-language and language-to-trajectory generation tasks offers a novel approach to understanding and generating human driving behaviors. I like the idea of treating map and trajectory tokens as the same latent space as language.
2. By incorporating lightweight encoders and a small language model (TinyLlama), Bi-Gen achieves annotation performance comparable to larger models like GPT-4o while remaining computationally efficient and suitable for real-time applications.
3. Flexible Multi-turn Interaction: The model’s multi-turn question-answering framework supports dynamic, interactive annotations and diverse trajectory generation, demonstrating versatility in handling complex driving scenarios.

**Weaknesses:**

1. The experiments focus on a racing domain with specific trajectory types, which may not generalize well to broader driving scenarios or other real-world applications without additional testing. I want to know if it's possible to extend to multi-agent scenarios, for example Waymo or Nuplan scenarios.
2. While the use of lightweight encoders and TinyLlama enhances efficiency, it might limit the model's capacity to capture finer details in complex, multi-modal interactions compared to larger models.
3. Bi-Gen’s performance relies on well-defined question-answer and generation prompts, which may limit its adaptability to novel or unexpected queries in deployment.
4. The paper does not explore the impact of different architectural choices (e.g., encoder and decoder sizes, tokenization approaches), which would strengthen understanding of the model's design trade-offs.
5. Following point 4, the tokenization approaches to the map and trajectory are unclear. See questions.

**Questions:**

1. How well does Bi-Gen generalize to other driving domains beyond multi-car racing? Could the model effectively handle scenarios with more varied driving behaviors, such as urban or highway driving in Waymo dataset?
2. How does Bi-Gen handle instances where trajectory descriptions or generation prompts are ambiguous or open-ended?
3. How the map and trajectory being tokenized? Are you using global coordinates? Do you do any transformation on the input data? In multi-agent complex scenarios such as Waymo dataset, there are huge number of elements in the scenes: multiple agents with different types and rich features (shape, velocity, type  etc), map features with different types (stop sign, different types of lanes etc), and even traffic light. So I am very concerning about whether the proposed method can be extended to other data format.

---

### Note · Authors · 2024-11-27

I have read and agree with the venue's withdrawal policy on behalf of myself and my co-authors.